# A Case Report of a Patient on Therapeutic Warfarin Who Died of COVID-19 Infection with a Sudden Rise in D-Dimer

**DOI:** 10.3390/biomedicines9101382

**Published:** 2021-10-03

**Authors:** Reita N. Agarwal, Hersheth Aggarwal, Ashmit Verma, Manish K. Tripathi

**Affiliations:** 1Department of Internal Medicine, Memphis VA Hospital, Memphis, TN 38104, USA; 2Health Science Center, College of Medicine, The University of Tennessee, Memphis, TN 38104, USA; haggarwal@uthsc.edu; 3Department of Biomedical Engineering, Samrat Ashok Technological Institute, Vidisha 464001, India; ashmitverma1998@gmail.com; 4South Texas Center of Excellence in Cancer Research, School of Medicine, University of Texas Rio Grande Valley, McAllen, TX 78504, USA; 5Department of Immunology and Microbiology, School of Medicine, University of Texas Rio Grande Valley, McAllen, TX 78504, USA

**Keywords:** COVID-19, warfarin, heparin, coagulopathy

## Abstract

Severe acute respiratory syndrome coronavirus 2 (SARS-CoV-2) infection has disrupted social and economic life globally. The global pandemic COVID-19 caused by this novel SARS-CoV-2 shows variable clinical manifestations, complicated further by cytokine storm, co-infections, and coagulopathy, leading to severe cases and death. Thrombotic complications arise due to complex and unique interplay between coronaviruses and host cells, inflammatory response, and the coagulation system. Heparin and derivatives are World Health Organization (WHO) recommended anticoagulants for moderate and severe Corona Virus Disease 19 (COVID-19), that can also inhibit viral adhesion to the cell membrane by interfering with heparan sulfate-dependent binding to angiotensin-converting enzyme 2 (ACE2) receptor. Heparin also possesses anti-inflammatory, immunomodulatory, antiviral, and anti-complement activity, which offers a benefit in limiting viral and microbial infectivity and anticoagulation from the immune-thrombosis system. Here we present a case study of the pathophysiology of unexpected COVID-19 coagulopathy of an obese African American patient. While being on therapeutic warfarin since admission, he had a dismal outcome due to cardio-pulmonary arrest after the sudden rise in D-dimer value from 1.1 to >20. This indicates that for such patients on chronic warfarin anticoagulation with “moderate COVID 19 syndromes”, warfarin anticoagulation may not be suitable compared to heparin and its derivatives. Further research should be done to understand the beneficial role of heparin and its derivatives compared to warfarin for COVID-19 inflicted patients.

## 1. Introduction

Today, SARS-CoV-2 infection has caused a COVID-19 pandemic worldwide, disrupting our society’s social and economic aspects [1]. There have been more than 2.83 million deaths worldwide. This is a challenging virus with unique transmission and infectivity properties [2].

SARS-CoV-2 infection is responsible for variable clinical manifestations, ranging from no symptoms to severe pneumonia with acute respiratory distress syndrome, septic shock, and multi-organ failure resulting in death [3]. Current strategies like isolation and social distancing are being used to reduce the need for hospitalizations. Treatment of the disease mainly focuses on symptomatic treatment and supportive care, including treatment of potential co-infection agents and early anticoagulation. Moreover, there are no treatments of absolute proven efficacy to reduce the progression of the disease from mild/moderate to severe/critical [4].

One of the most important mechanisms underlying the deterioration of disease is the cytokine storm [5], where elevated levels of pro-inflammatory molecules, such as interferons a and b, TNF alpha and interleukin-4 (IL-4), interleukin-6 (IL-6), and interleukin-8 (IL-8) [6] are seen. The latest trial has shown promise for remdesivir, lopinavir-ritonavir, dexamethasone, methylprednisolone, IL-6 inhibitors, monoclonal antibodies, and convalescent plasma, to name a few [7]. Laboratory findings of COVID-19 include lymphopenia with depletion of cluster of differentiation 4 (CD4) and cluster of differentiation 8 (CD8) lymphocytes, prolonged prothrombin time (PT), elevated lactate dehydrogenase (LDH), D-dimer, alanine transaminase (ALT), C-reactive protein (CRP), and creatinine kinase (CK) [8,9].

Severe disease is also complicated by coagulopathy and disseminated intravascular coagulation (DIC) with a very high risk of death [10]. However, inflammation and coagulation are linked. The innate immune defense and coagulation system, collectively known as the immune-thrombosis system, produces a distinct intravascular thrombi compartment early on to limit microbe insult on tissues and allow rapid healing. Insufficient coagulation can result in bleeding and hemorrhage, while excessive clotting may lead to thrombosis or even disseminated intravascular coagulation (DIC). Patients with progressive, severe COVID-19 infection with acute lung injury or acute respiratory distress syndrome (ARDS) have very high D-dimer levels and low fibrinogen levels because of consumptive coagulopathy. Therefore, the use of anticoagulants for patients with severe COVID-19 has been recommended by expert consensus and the World Health Organization (WHO) [11,12,13]. There is more than ample existing evidence on the use of heparin and related derivatives, including fractionated heparin, low-molecular-weight heparin (LMWH), and direct oral anticoagulants (DOACs), to prevent or treat thrombotic complications in moderate to severe COVID-19 cases. This therapy balances the drug–drug interaction and individual risk of thrombosis versus bleeding [14,15].

Recent studies have shown that the role of heparin in COVID-19 is more than just anticoagulation. Studies have described its “pleiotropic activity”, but this must be further proven in clinical trials [16]. Molecular modeling studies show that the SARS-CoV-2 spike protein interacts with both host cellular receptor heparan sulfate and ACE2 through its receptor-binding domain (RBD) [17]. Hence in a model in which viral attachment and infection involves heparan sulfate-dependent enhancement of binding to the ACE2 receptor (Figure 1), then exogenous heparin will not only be part of anticoagulation but also inhibit viral adhesion to the cell membrane, affecting infectivity. This mechanism of infection could reveal other targets to interfere with viral infection and spread.

## 2. Case Presentation

In our case observation, we had a 71-year-old obese (body mass index (BMI) 33, weight 204.8 pounds/92.9 kg, height 66 inches/167.64 cm) African American male who looked younger than his age with a relevant past medical history (Appendix A, Past medical history for details and Appendix A for Home medication list) of uncontrolled hypertension but controlled hyperlipidemia, compensated Class IA systolic cardiomyopathy with ejection fraction 35–40%, coronary heart disease, diabetes hemoglobinA1c (HgA1c 8.3%), pulmonary embolism (PE) from provoked leg injury-related deep vein thrombosis (DVT) on warfarin International Normalized Ratio (INR) 2–3 due to recurrence in 2015, pelvo-abdominal horse-shoe kidney, Gold class 1A chronic obstructive pulmonary disease (COPD), fatty liver, mild anemia, tobacco use in remission and intermittent alcohol. He presented with 4–6 days of subjective fever, and worsening sinus and cough symptoms causing dyspnea and dizziness with mild exertion from walking to the bathroom from bed. He also had generalized weakness and cough with clear to yellow phlegm. He had no chest pain, orthopnea, or paroxysmal nocturnal dyspnea but had palpitations on mild exertion. Patient denied nausea, vomiting, diarrhea, dysuria or leg swelling. On arrival, he was alert and oriented. His significant vital signs at rest (Appendix A) were afebrile, with a heart rate of 87–98 and BP (blood pressure) of 201/110–140/92, and RR (respiratory rate) of 14–18. He was hypoxic on room air with O_2_ sats (oxygen saturation) of 93% at room air, but when he stood up to go to the bathroom, his O_2_ sats dropped to 85% on room air, RR rose to 24, HR rose to 120, and SBP (systolic blood pressure) dropped to 96. He was stabilized and required 2–3 L of oxygen saturation with 94–96% and intravenous normal saline fluids for hypotensive SBP. He also felt dizzy getting up. On exam, he was feeling weak but alert and oriented to name, place and date. He was ambulatory, but only for a very short distance before he would decompensate, as mentioned above. He had rhonchi in both lungs diffusely with no other significant findings on other organ systems. Significant negative exam findings were no jugular venous pressure (JVP), heart murmur, or leg edema. His significant bloodwork showed he was positive for the Covid-19 molecular test, elevated liver function tests (LFT) (Table 1), elevated COVID inflammatory panel with normal pro-BNP (brain natriuretic peptide) and troponin-I (Table 2), urine showing glucosuria and a high specific gravity of 1.024. A low vitamin D of 11 and folate of 5.2 were noted (Table 3). His prothrombin time and international normalized ratio (PT/INR) was therapeutic at 2.5 with elevated platelets at 480/uL, and mild, stable anemia (Table 4). Chest X-ray and CT angiography (CTA) showed no pulmonary embolism (PE) but bilateral multilobe ground-glass opacities (GGO). His electrocardiogram (EKG) was sinus tachycardia with no ST-T changes. Table 5 presents the blood chemistry. On admission, the patient received fluids and methylprednisolone (80 mg drip), and remdesivir was started the following day along with community bacterial pneumonia antibiotics (ceftriaxone 2 g with azithromycin 500 mg) and mometasone inhaler. He received COVID-19 related supplements including vitamin C, vitamin D, zinc, melatonin, thiamine, and folate as part of his inpatient medications (Appendix A). He declined experimental treatment. He repeatedly declined consent for convalescent plasma. He was generally improving as perceived by the patient symptomatically and based on his inflammatory markers (Table 4) and other labs, except for platelet count, which was high on admission but dropped to normal on day 4 (Table 1). He did not require supplemental oxygen at rest or walking short distances within the room but it was still required for walking during physical therapy by day 3. He expressed the desire to go home on oxygen as needed when remdesivir infusion was completed. Unfortunately, on Day 4, he decompensated suddenly late afternoon with a mild rise in troponin-I (0.06) and a D-Dimer rise to >20 from 1.1 (Table 4), while being therapeutic on warfarin since admission. His anemia was stable although his white blood cells (WBC) rose (Table 1). We attributed this to methylprednisolone. His cultures showed no growth in both blood and urine. His chest X-ray from late the night before showed worsening of infiltrates. His oxygen requirement increased from 2 L to 4 L later on the day of decompensation. The patient did not believe our concerns of an impending critical adverse event. After educating the patient in the presence of our African American respiratory therapist, we immediately ordered the conversion of warfarin to therapeutic Lovenox in addition to EKG, telemetry, and CTA for PE. Unfortunately, we were too late; while going for CTA, he died from cardio-pulmonary arrest with initial supraventricular tachycardia (SVT) with a rapid ventricular response (RVR) and then pulseless electrical activity (PEA) that could not be reversed. CPR (cardio-pulmonary resuscitation) teams documented right heart strain on bedside echo when in PEA. Unfortunately, the family declined postmortem, but we were able to get levels of IL-4, IL-8 and confirmed COVID-19 positive antibodies (Table 6) from a small tube of blood left from that day. There was not enough blood for other studies.

## 3. Discussion

In our patient, it is concerning that while inflammatory markers were improving, the patient had suddenly developed a D-dimer of greater than 20 despite having therapeutic levels of warfarin. We also noted a minor rise in troponin-I (Table 4). It was interesting to see that IL-4 was not detected while IL-8 was very high. Noted SARS-coV-2 total antibodies were elevated but less than 300 (Table 6). Coagulopathy due to progressive cytokine storm is a significant complication of COVID-19 and needs to be studied further for treatment modification. Troponin rise seems to be due to type 2 ischemia related to right heart strain from hyper-coagulopathy. It was surprising to see that IL-4 was undetected. In the literature, IL-4 has been shown to increase in COVID-19 patients [18], unless the patient had severe lung injury due to depletion of innate antiviral immune response defenses. When the D-dimer rose to >20, the patient seemed to have lost all immune defenses. However, the patient was not in disseminated intravascular coagulation (DIC) based on stable hematocrit and INR. Very high IL-8 was expected with disease progression with the rise in SARS-CoV-2 antibodies. The level of antibodies <300 indicated that immunity was not yet fully developed. This case indicates that warfarin may not be an effective treatment for COVID-19 related coagulopathy. Instead, heparin and its derivates might be more efficacious alternatives. This was the only patient we did not give heparin products to on admission, who died unexpectedly thinking he was anticoagulated. The inability of therapeutic warfarin anticoagulation to control the progression of COVID-19 hypercoagulation in moderate COVID 19 syndromes is hidden in the uniqueness of heparin and its relationship to the immune-thrombosis system.

Inflammation and coagulation are linked. The coagulation pathway (Figure 1) is a cascade of events that leads to hemostasis and allows rapid healing. Acute infections, including viral ones, induce host cell/tissue disruption leading to systemic inflammatory response and coagulation disruption [19]. Tissue injury leads to a decrease in antithrombin (AT), which potentiates the coagulation process. Both the coagulation pathways, intrinsic and extrinsic, come together at factor Xa in a common pathway (Figure 1). The innate immune defense and coagulation system, collectively known as the immune-thrombosis system, produces a distinct intravascular thrombi compartment with antimicrobial peptides to limit microbial dissemination [20,21]. Heparin and its derivatives are involved in the immune-thrombosis system in containing early infection [17,22].

Heparin and warfarin characteristics are summarized in Figure 1. As an anticoagulant, heparin and its derivatives bind to ATIII, which inactivates factor Xa and IIa, affecting the intrinsic pathway, measured as prolonged PTT. The anticoagulant activity of heparin is associated with the formation of the heparin-antithrombin complex. Antithrombin is a plasma α2 globulin that inhibits the activity of coagulation factors, which are serine proteases. The complex of antithrombin with heparin increases the inhibitory effect of antithrombin on coagulation factors by up to 1000-fold. The heparin-antithrombin complex inhibits enzymatic activity and neutralizes the active form of factors X and XII, XI, IX, and thrombin. In the case of thrombin, the effect of the heparin-antithrombin complex is mainly due to thrombin formation rather than inhibition of thrombin activity. High concentrations of heparin may also inhibit platelet aggregation [23]. At the same time, warfarin inhibits the synthesis of Vit K-dependent factors (factors II, VII, IX, X, protein C and S), which slows the body’s ability to form clots, affecting the extrinsic pathway, measured as prolonged PT/INR. Warfarin, a vitamin K antagonist anticoagulant drug, belongs to the 4-hydroxycoumarins like acenocoumarol. The primary mechanism of the anticoagulant activity of 4-hydroxycoumarins is the inhibition of vitamin K epoxide reductase. This enzyme restores the active form of vitamin K by converting the inactive alkoxide-epoxide form of vitamin K into its active reduced hydroquinone form. Reduced vitamin K is an essential cofactor of hepatic γ-glutamyl carboxylase activity. This enzyme adds a carboxyl group to the glutamic acid residues in immature clotting factors II, VII, IX, and X as well as proteins S, C, and Z. Inhibition of γ-glutamyl carboxylase activity due to a deficiency of reduced vitamin K leads to the formation of immature forms of coagulation factors that cannot be converted to the active forms. This leads to a reduction in blood clotting [24,25]. Studies have shown that both heparin [26] and coumarins such as acenocoumarol [24,27,28] and warfarin [25,29] have anti-inflammatory and therapeutic effects in experimental pancreatitis. This observation suggests that in other diseases with an inflammatory component, heparin and warfarin may be helpful in the treatment of these diseases.

The role of heparin in COVID-19 infection is complex. In the form of heparan sulfate proteoglycans (HSPGs), heparin has the added benefit of being host cell surface receptors for pathologic proteins and viruses [30]. Hence, heparin is involved in “pleiotropic activity”, which is anti-inflammatory, immunomodulatory, antiviral, and anti-complement [16]. Interestingly, experimental studies have shown that the SARS-CoV-2 spike protein interacts with not just ACE2; but also cellular heparan sulfate proteoglycans (HSPGs) through its RBD (receptor binding domain) [12]. Hence, both ACE2 and HSPG receptors on the cell surface are needed for the SARS-CoV-2 virus to enter the target host cell. Thus, heparin and heparan sulfate can antagonize the binding of pathogens to HSPGs on host cells and stop their cellular internalization and dissemination, acting as a therapeutic armamentarium against COVID-19 in addition to the anticoagulation effect [17,31]. A model representing viral replication with and without heparin is depicted in Figure 2.

## 4. Conclusions

The patient’s dismal outcome due to progression of moderate COVID-19 to sudden severe and then critical COVID-19 status while on therapeutic warfarin indicates that alternatives to warfarin anticoagulation such as heparin and related derivatives should be considered. Since the patient was not on heparin, we hypothesize that continued viral cellular internalization through HSPGs, and ACE2 receptors bound to related COVID-19 spike proteins leads to virus dissemination and increasing host viral infectivity. Unlike warfarin, studies have shown that heparin possesses anti-inflammatory, immunomodulatory, antiviral, and anti-complement activity against microbial injury, which could have benefited from limiting viral and microbial infectivity from immune-thrombosis and anticoagulation. Further studies should be done comparing the efficacy of warfarin versus heparin in COVID-19. The patient had COVID 19 antibodies and elevated IL-8, but no IL4 was detected. Coagulopathy due to the progressive cytokine storm of COVID-19 needs to be studied further for treatment modification on therapeutic warfarin pre-COVID 19 infections. Further studies are also needed on HSPGs affecting COVID-19 virus entry, inflammation, immune activation, and the pleiotropic role of heparin in COVID-19 to reduce host viral infectivity.

Heparin and heparan sulfate antagonizes the binding of these pathogens to HSPGs and stops their cellular internalization. However, the anticoagulant effect of these agents has been limiting their use in the treatment of viral infections. Perhaps heparin-binding peptides (HBPs) are suitable nonanticoagulated agents that may be capable of antagonizing binding of heparin-binding pathogens (HBP) to HSPGs. Thus, the use of HBPs as viral uptake inhibitors may address their benefits and limitations to treat viral infections. Furthermore, a variant of these peptides can be considered a novel therapy in coronavirus disease 2019 (COVID-19) infection [16,22]. Recently, a retrospective cohort study analyzed the relieving effect of LMWH in patients with COVID-19 to investigate the anti-inflammatory effects of heparin and the delay of disease progression [32]. Compared to the control group, patients treated with heparin improved hypercoagulability, and showed a reduction in IL-6 and neutralization of its biological activity, and an increase in the percentage of lymphocytes. However, the same benefit and safety of heparin as anti-inflammatory and antiviral agents in a clinical setting are yet to be defined due to conflicting results reported by previous clinical trials.

## 5. Clinical Relevance

Our observation of a patient on therapeutic warfarin, despite high dose steroids and remdesivir, progressed suddenly to death with a sudden rise in D-dimer greater than 20. For such patients who are on chronic warfarin anticoagulation, when they are diagnosed with “moderate COVID 19 syndrome”, warfarin anticoagulation may not be as suitable compared to heparin and its derivatives. In addition, D-dimer comes from both intravascular and extravascular fibrin. It is known that this test is characterized by a high sensitivity and low specificity [33]. This may happen in patients with nCoV-19 infection who suffer from pneumonitis, which can be the source of D-dimer from extensive alveolar fibrin deposition [34]. However, this patient had low IL-4 [21].

Based on the studies mentioned above, since the patient was not on heparin-related anticoagulation, we hypothesize that continued viral uptake and cellular internalization through the patient’s HSPGs related COVID-19 receptors lead to dissemination of virus and increasing host viral infectivity. Unlike warfarin, studies have shown that heparin possesses anti-inflammatory, immunomodulatory, antiviral, and anti-complement activity, which could have offered a benefit in limiting viral and microbial infectivity along with anticoagulation to inhibit the progression of moderate COVID 19 syndromes. Unfortunately, this patient did not reduce infectivity, leading to worsening cytokine storm and hemostasis. The patient had COVID 19 antibodies and elevated IL-8, but no IL-4 was detected. We did not have enough blood to check other cytokines. The family opted not to get an autopsy. Inflammatory and angiogenic biomarker IL-8, similar to IL-6 but with a more extended life, is a chemotactic factor that attracts neutrophils, basophils, and T-cells during the inflammatory process. It is close to platelet factor 4.

While serum IL-6 becomes elevated in severe COVID-19 patients, serum IL-8 was quickly detectable in COVID-19 patients with mild syndromes and severe disease [20]. In addition, various studies of COVID-19 patients have detected elevated IL-4 levels as part of the cytokine storm associated with severe respiratory symptoms [21], which is also puzzling.

Further studies should be done comparing the efficacy of warfarin versus heparin in moderate COVID-19. Data analysis is critical to compare COVID patients on chronic warfarin and other anticoagulants. Perhaps a pilot should be done for such patients on warfarin with and without heparin-based therapeutic anticoagulation. Fondaparinux properties are equally crucial as heparin derivatives [35] and can also be studied as AT III inhibitors. There may be genetic and race factors contributing to this coagulopathy. We wondered if horse-shoe kidney had any role to play. It is true that neither heparins nor warfarin may be able to reverse this severe “cytokine’ storm”. It is also known that the course of the COVID-19 disease can suddenly be worse. Either way, based on drawbacks of warfarin [36] compared to the advantages of using heparin and its derivatives, we strongly recommend further conclusive research for considering a change in the treatment protocol for COVID 19 syndrome patients on chronic warfarin to be converted to therapeutic anticoagulation with heparin and its derivatives for improved survival outcome.

## Figures and Tables

**Figure 1 biomedicines-09-01382-f001:**
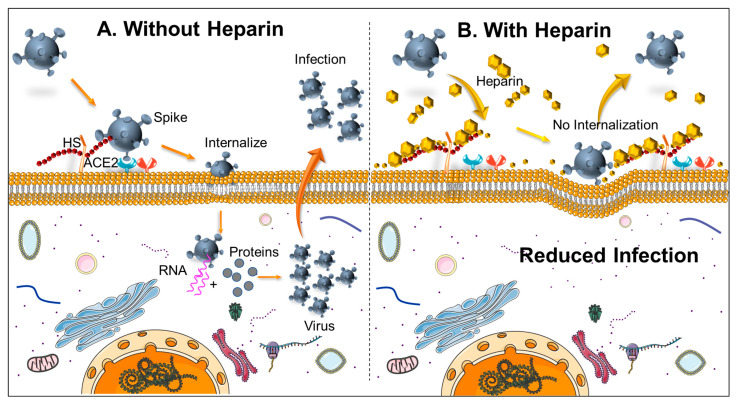
Schematic showing (**A**) In the absence of Heparin, SARS-CoV-2 receptor binding domain of spike glycoprotein interacts with heparan sulfate (HS) and can facilitate virus internalization through ACE2. As a result, viral components replicate inside the cell and finally are released to increase the infection to the neighboring cell, (**B**) in the presence of heparin, the accessibility to heparin sulfate proteoglycan is decreased, and hence the virus is not able to internalize, which results in a decrease in virus binding to the cell surface.

**Figure 2 biomedicines-09-01382-f002:**
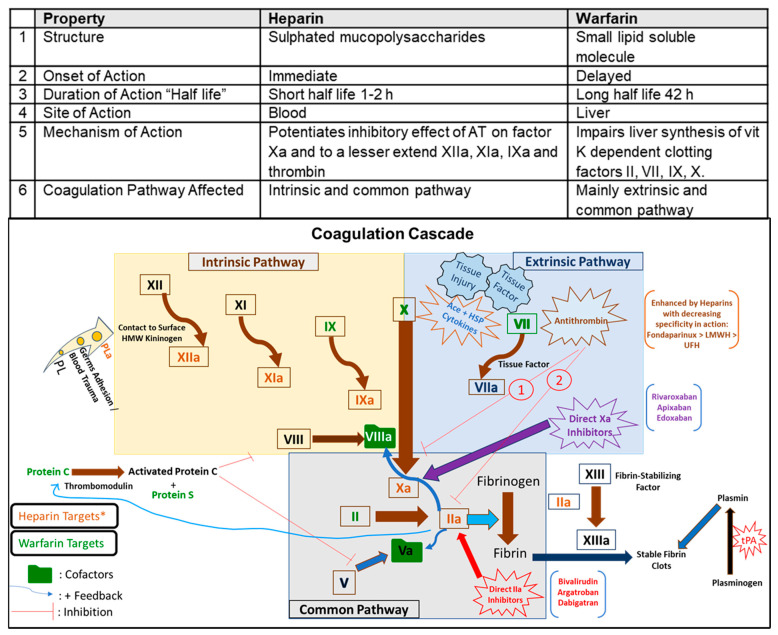
Blood coagulation pathways and comparison of the mechanisms of the anticoagulant activity of heparin and warfarin. vitK: Vitamin K; PL: Platelets; tPA: tissue Plasminogen activator; HMW: High Molecular Weight; PL: Phospholipid; PLa: activated Phospholipid; HSP: Heparan Sulphate Polysaccharides; LMWH: Low molecular weight heparin; UFH: Unfractionated heparin, “a” next to the names: activated form, AT: Antithrombin.

**Table 1 biomedicines-09-01382-t001:** Liver function profile.

Marker	Units	Day 0/Admit	Day 1	Day 2	Day 3	Day 4/Death
Protein	g/dL	7.1	6.7	6.0 L	6.1 L	
Albumin	g/dL	3.2 L	3.1 L	2.7 L	2.8 L	
T.Bil	mg/dL	1.1	1.1	0.5	0.6	
AlkPhos	U/L	68	67	68	77	
A	U/L	75 H	62 H	67 H	35 H	
ALT01	U/L	59 H	53	61 H	50	

Tbil: Total Bilirubin; AlkPhos: Alkaline Phosphatase; AST01: Aspartate Aminotransferase; ALT01: Alanine Aminotransferase.

**Table 2 biomedicines-09-01382-t002:** Inflammatory labs profile.

Marker	Units	Day 0/Admit	Day 1	Day 2	Day 3	Day 4/Death
PT	Sec		26.7 H	27.5 H	26.1 H	26.4 H
INR	Ratio		2.50 H	2.60 H	2.43 H	2.48
Fibrinogen	mg/dL		818 H			
D-Dimer	ug/mL		1.14 H	1.04 H	-	>20.0 H
CRP	mg/L		126.6 H	81.8 H	-	60.7 H
Ferritin	ng/mL		2495.0 H	2432.0 H	-	1845.0 H
Procalcitonin	ng/mL		0.14	0.09	-	0.06
LDH-V	U/L		650 H	646 H	-	780 H
Lactic Acid	mmol/L		1.5	1.5	-	1.5
CPK	U/L		496 H	318 H	-	165
Troponin-I	ng/mL	0.0	0.028	-	-	0.063 H
BNP	pg/mL		58	50		36
Cortisol	ug/dL			2.6		

PT: Prothrombin time; INR: International Normalized Ratio; F-Xa: Factor Xa; CRP: C-Reactive Protein; LDH-V: Lactate Dehydrogenase; CPK: Creatine Phospho-Kinase; BNP: Brain Natriuretic Peptide.

**Table 3 biomedicines-09-01382-t003:** Preventive labs profile.

Marker	Units	Day 0/Admit	Day 1	Day 2
HgbA1c	%			8.3 H
Chol	mg/dL			138
Trigly	mg/dL			84
HDL	mg/dL			39
LDL	mg/dL			82.2
TSH (0)	ulU/mL			0.6
VitD	ng/mL			11
B12 (0)	pg/mL			379
FolA (0)	ng/mL			5.2 L
Iron	ug/dL			34 L

HgbA1c: Hemoglobin A1c; Chol: Cholesterol; TriGly: Triglycerides; HDL: High-Density Lipoprotein; LDL: Low-Density Lipoprotein; TSH: Thyroid Stimulating Hormone; VitD: Vitamin D; B12: Vitamin B12; FOLA: Folic Acid.

**Table 4 biomedicines-09-01382-t004:** Complete blood count (CBC) profile.

CBC	Units	Day 0/Admit	Day 1	Day 2	Day 3	Day 4/Death
WBC4	10^3^/uL	10.66 H	8.96 H	12.29 H	11.43 H	11.61 H
HGB4	g/dL	12. L	11.6 L	10.6 L	10.9 L	10.60 L
HCT4	%	35.0 L	32.9 L	29.9 L	30.7 L	29.8 L
PLT4	10^3^/uL	480.0 H	468.0 H	517.0 H	544.0 H	293 H
NEUT%4	%	87.5 H	78.4 H	86.9 H	85.7 H	80.0 H
LY%4	%	7.5 L	15.4 L	6.8 L	7.2 L	8.9 L

WBC4: white blood cells; HGB4: Hemoglobin; HCT4: Hematocrit; PLT4: Platelets; NEUT%4: Neutrophils; LY%4: Lymphocyte.

**Table 5 biomedicines-09-01382-t005:** Blood chemistry profile.

Chemistry	Units	Day 0/Admit	Day 1	Day 2	Day 3	Day 4/death
Na	mmol/L	137	139	133 L	136	136
K^+^	mmol/L	3.3 L	3.2 L	4	4	3.6
CL^−^	mmol/L	97 L	99 L	101	104	103
CO_2_-V	mmol/L	29 H	27	22	23	21 L
BUN	mg/dL	15	17	26 H	19	19
Creat2	mg/dL	0.96	0.97	1.12	0.96	0.83
Gluc	mg/dL	133 H	139 H	323 H	242 H	236 H
Ca	mg/dL	8.3 L	8.2 L	7.8 L	8.0 L	8.0 L
iCa	nmol/L		0.99 L	1.01 L	1.05 L	1.14 L
Phos	mg/dL		1.5 L	2.9	2.2 L	2.6
Mg^++^	mg/dL		1.6	2.8	1.9	1.6
EGFR	mL/min		78	78.2	93.4	110.5

Na: Sodium; K^+^: Potassium; Cl^−^: Chloride; CO_2_-V: Carbon dioxide/Bicarbonate; BUN: Blood urea nitrogen; CREAT2: Creatine; GLUC: Glucose; CA: calcium; iCA: ionized Calcium; Phos: Phosphate; Mg^++^: Magnesium; eGFR: estimated glomerular filtration rate.

**Table 6 biomedicines-09-01382-t006:** Interleukin and antibody tests after death.

Tests	Units	Result	Flag	Ref. Intervals
IL-4, Serum	pg/mL	<31.2		0.0–31.2
IL-8, Serum	U/mL	437.1	HIGH	0.0–66.1
SARS-CoV-2 Semi-Quant Total Ab	U/mL	224.9	HIGH	<0.80

IL-4: interleukin-4; IL-8: Interleukin-8; Ab: Antibody.

## Data Availability

Not applicable.

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
