# Peer review of "A Case Report of a Patient on Therapeutic Warfarin Who Died of COVID-19 Infection with a Sudden Rise in D-Dimer"

_biomedicines, 2021, doi:10.3390/biomedicines9101382_

Round 1

Reviewer 1 Report

Manuscript ID biomedicines-1378927 (previous submission biomedicines-1302729)

Title: Therapeutic Warfarin instead of Heparin & derivatives may not improve Moderate Covid-19 Syndrome

Journal: Biomedicine

Authors: Agarwal et al.

The manuscript is a case report presenting the course COVID-19 infection in a patients treated with warfarin due to risk of deep vein thrombosis in the lower limb. The manuscript is interesting, and its current version is much better than the previous one. In the new version of the manuscript, the authors have presented all relevant data on the patient’s condition before and during hospitalization. On the other hand, there are some remarks regarding the relationship between inflammation and clotting, as well as the therapeutic potential of anticoagulants that should be presented in the manuscript before its acceptance for publication.

List of major deficiencies and errors:

  1. The title of the study. The previous title “Therapeutic Warfarin May Worsen Moderate COVID-19 Syndrome: A Case Report” was pure speculation. The authors did not provide any evidence that warfarin exacerbates the course of the coronavirus infection. The average mortality from COVID-19 infection is around 2%, depending on the type of virus, patient age and comorbidities, and there is no treatment that cures all patients. Moreover, all humans are mortal and must die. The question is only about the time of death and its cause. Thus, the death of a patient with coronavirus infection and taking warfarin does give a reason to conclude that warfarin worsened the course of the coronavirus infection. The new title is better, but still does not refer to the authors’ observations. The authors did not conduct clinical trials comparing the therapeutic effect of heparin and warfarin administration in the course of COVID-19 infection. Authors should consider a more neutral title such as, for example, “Case report of a warfarin-treated patient who died of COVID-19 infection. The interaction between clot activation and inflammation”. Or something like that.
  2. Introduction, a new paragraph 3. The authors should consider adding a new paragraph regarding the role of coagulation in the homeostasis. The authors should write that coagulation prevents blood loss after tissue and vessel damage. However, as with other homeostatic mechanisms, clotting activity should be kept within an adequate range. Insufficient coagulation can result in bleeding and hemorrhage, while excessive clotting may lead to thrombosis or even disseminated intravascular coagulation (DIC).
  3. Introduction. The authors refer to Figure 1 in the text (page 2, line 73) but the text actually refers to Figure 2. There is no real reference in the text to Figure 1. Therefore, the authors should add the text relating to the activation of coagulation in intrinsic, extrinsic and common pathway. The authors should present describe the mechanism of activation of intrinsic and extrinsic pathways, as well as write that both pathways, intrinsic and extrinsic lead to activation of final common pathway with creation of active factor X (Xa), thrombin and fibrin.
  4. In addition, the authors should perform some correction of Figure 1. Row 3: Duration of Action (T-half life), change the name of the row into ”Half life”. In the second column in this row, write “Short half life: 1-2 h”. In the third column of this row write “Long half life 20-60 h”. The abbreviation of hours is h.
  5. Figure 1. Row 5: “Mechanism of action”. In the second column in this row, write “Potentiates inhibitory effect of AT on factor Xa and to a lesser extend XIIa, XIa, IXa and thrombin”. Factor IIa and thrombin are the same.
  6. Figure 1. Row 6 “Coagulation Pathway Affected”. In the second column of this row write “Intrinsic and common pathway”. In the third column write “Mainly extrinsic and common pathway”.
  7. Figure 1. The diagram in Figure 1. Add an arrow connecting Factor IIa (thrombin) with an arrow showing conversion of fibrinogen to fibrin. Alternatively, the symbol IIa can be moved towards this second arrow.
  8. Figure legend of Figure 1. Change the figure legend to “Blood coagulation pathways and comparison of the mechanisms of anticoagulant activity of heparin and warfarin”. In addition, the authors should leave the list of abbreviation and add the abbreviation AT to the list. Authors should also know that antithrombin III is now called antithrombin and that last name should be used throughout all parts of the manuscript.
  9. Introduction. In addition, in the text relating to Figure 1, the authors should describe anticoagulative mechanisms and effects of heparin and vitamin K antagonist. In the case of heparin, the authors can write that the anticoagulant activity of heparin is associated with formation of the heparin-antithrombin complex. Antithrombin is a plasma α2 globulin that inhibits the activity of coagulation factors which are serine proteases. The complex of antithrombin with heparin increases the inhibitory effect of antithrombin on coagulation factors by up 1000-fold. The heparin-antithrombin complex inhibits enzymatic activity and neutralizes the active form of factor X, as well as XII, XI, IX and thrombin. In the case of thrombin, the effect of the heparin-antithrombin complex is mainly due to of thrombin formation rather than inhibition of thrombin activity. Moreover, high concentrations of heparin may also inhibit platelet aggregation (PMID: 18955758).
  10. Introduction. In the text relating to Figure 1 and regarding the anticoagulant activity of warfarin, the authors should state that warfarin, like acenocoumarol, belongs to the 4-hydroxycoumarins, a vitamin K antagonist anticoagulant drugs. The primary mechanism of anticoagulant activity of 4-hydroxycoumarins is inhibition of vitamin K epoxide reductase, an enzyme that restores the active form of vitamin K by converting the inactive alkoxide-epoxide form of vitamin K into its active reduced hydroquinone form. Reduced vitamin K is an essential cofactor of the hepatic γ-glutamyl carboxylase activity. This enzyme adds a carboxyl group to the glutamic acid residues in immature clotting factors II, VII, IX, and X as well as proteins S, C and Z. Inhibition of γ-glutamyl carboxylase activity due to a deficiency of reduced vitamin K leads to the formation of immature forms of coagulation factors that cannot be converted to the active forms. This leads to a reduction in blood clotting (PMID: 27754317; PMID: 32471279).
  11. Discussion. The authors should describe the interaction between coagulation, and inflammation. They should write that previous studies have shown that there is a close relationship between clotting and the development of inflammation. This relationship is two-sided. Coagulation stimulates the development of inflammation, and at the same time, inflammation activates the coagulation cascade (PMID: 19437587; PMID: 23955016). The authors should also present factors involved in pro-inflammatory effects of coagulation products and mechanisms of this activity, as well as pro-coagulative effects of inflammation using data presented in article PMID: 33077694.
  12. Discussion. The authors should mention in the discussion that studies on experimental acute pancreatitis have shown that both heparin (PMID: 23070084) and coumarins such acenocoumarol (PMID: 26579579; PMID: 27754317; PMID: 28430136) and warfarin (PMID: 32471279; PMID: 33077694) have anti-inflammatory and therapeutic effects in this disease. This observation suggests that also in other diseases with an inflammatory component, heparin and warfarin may be useful in the treatment of these diseases. 

List of minor deficiencies and errors:

  1. All abbreviations should be presented in full form at the place where they are used for the first time in the abstract and repeated in the main body of the manuscript. Figure legends and Tables.
  2. For each drug, the authors should provide its generic name, trade name, and name of manufacturer, city and country (see for example line 137).
  3. Abstract, line 18.  In “leading to death” between Leading” and “to death” add “in the severe cases”.
  4. Introduction, page 2, line 53. The authors should provide information on which classes of lymphocytes show the presence of CD4 and CD8 antigens.
  5. Introduction, page 2, line 57. Replace “leading to deaths” with “with very high risk of death”.
  6. Introduction, page 2, line 59. Replace “related to a hypercoagulative state” with “because of consumptive coagulopathy”.
  7. Case presentation. Many words are capitalized for unknown reason (for example line 85 - Height; line 88 – Hypertension; line 90 – Pulmonary; line 100 – Blood Pressure; line 116 – Bilateral). This should be corrected. In addition, a semicolon should be used between individual parameters instead a comma.
  8. Case presentation. Page 4, line 102. Change “o2” into “O2”.
  9. Case presentation, line 115. The authors should add units after the platelet count.
  10. The new version of the manuscript should be accompanied by responses to reviewer’s comments clearly describing where and what changes have been made to the manuscript.

Reviewer 2 Report

The revised form of this paper is satisfactory 

I have no other remarks. 

Round 2

Reviewer 1 Report

Manuscript ID biomedicines-1378927-peer-review-v2

The current title: A Case report of Patient on Therapeutic warfarin who died of COVID-19 infection with sudden rise in D-Dimer

Authors: Agarwal et al.

The manuscript is generally ready for printing. The reviewer has found only one important error on page 3, line 116. The authors reported that the patient had an increased platelet count and therefore it appears that the platelet count should be not 480/µl but 480,000/µl. This error can be corrected during proofreading, therefore the reviewer suggests accepting the manuscript for publication. On the other hand, congratulation to the authors for taking steps for applying “systemic review” of patients on chronic warfarin and covid-19, as a project that may lead to pilot clinical trial. Very good idea.

Author Response

This manuscript is a resubmission of an earlier submission. The following is a list of the peer review reports and author responses from that submission.

Round 1

Reviewer 1 Report

Manuscript ID biomedicines-1302729

Title: Therapeutic Warfarin May Worsen Moderate COVID-19 Syndrome: A Case Report

Journal: Biomedicines

Authors: Agarwal et al.

The manuscript is a case report presenting the course COVID-19 infection in the patients treated, inter alia with warfarin. The manuscript is interesting, but contains some deficiencies and errors that need to be corrected before the manuscript could be accepted for publication. All responses to the reviewer’s comment should be presented in the next version of the manuscript.

List of deficiencies and errors:

  1. All abbreviations should be presented in their full name at the point where they appear for the first time.
  2. Figures and their legends, as well as Tables must be understandable without references to the body of the manuscript. All abbreviations used in Figures and Tables should be presented in full name in figure legends, table headings or in table captions. In the case of numeric data of numeric data, the units in which these data are expressed should be provided. Metric units are preferred. Additionally, if there are the several possible forms of parameters tested, the authors should provide what exactly has been determined. It is especially important in the case of troponin and CPK.
  3. Section 2. Case Presentation. The authors should present more details concerning the patient before and during hospitalization. The authors should list all medication taken by the patient before admission to the hospital, as well as medication administered to the patients during hospitalization. This information should be presented in a separate Table. Generic drug names should be provided, as well as marked names. Additionally, the authors should provide medicine doses and frequency of administration.
  4. Section 2. Case Presentation. The authors should provide the patient’s body mass and height. Body mass index 33 has a very different meaning in short and tall patients. Body mass should be given in pounds and kilograms, and height in feet and inches, as well as in centimeters.
  5. Section 2. Case Presentation. The authors reported that the patients before hospitalization had fever. The authors should presented temperature reported by patients, as well as the patient’s body temperature during hospitalization.
  6. The authors reported that patient had the compensated Class IA systolic cardiomyopathy with ejection fraction 35-40%, coronary heart disease. When were these disorders diagnosed? Did the patients complain of any pains during admission to hospital and during hospital stay? If the patient was diagnosed with coronary heart disease before  hospitalization, it should be concluded that with generalized infection, hypoxia and tachycardia, there was a further reduction in the ejection fraction and an increase in myocardial hypoxia. The heart’s need for oxygen increased in direct proportion to the heart rate and the increase in cardiac output. The heart’s oxygen needs can only be met through an increase in coronary blood flow! Did the patient have echocardiography while in hospital? If so, when was it done and what were the results of this examination? Did the patient have an ECG? The authors reported that sinus tachycardia was found on the patient’s ECG, but it is unknown whether ECG was performed in the hospital, and if so, when? Were there any ECG changes showing coronary artery disease? If so, what was their nature? Has The ECG been repeated in any time? Troponin was not tested on day of admission to the hospital. On day 1 CPK was above normal value, however, again troponin testing was not performed. Why?
  7. Section 2. Case Presentation. The authors reported that the patient was “hypoxic on room air requiring 2-3 Liters of oxygen with saturation of 94-96%, hypotensive SBP 96”(lines 85-86). First, it should be assumed that the authors write about saturation of hemoglobin with oxygen. Was oxygen saturation 94-96 before or after the administration of oxygen? If, after, what was the initial oxygen saturation? How was oxygen saturation measured? Have the authors performed an arterial blood gas test? Were these measurements repeated? Second, what was the patient’s initial respiratory rate? Did it change during the stay in the hospital? Did the authors monitor patient’s arterial blood pressure (systolic and diastolic) oi the following days of hospitalization? All these data should be presented in the manuscript.
  8. Section 2. Case Presentation. Did the authors measure the patients water balance? At least, fluid intake and diuresis should be determined. Has there been fluid retention and hypervolemia? Decreased level of hematocrit, hemoglobin, protein, albumin and ASPAT during hospital may indicate hypervolemia. Were central venous pressure and real glomerular filtration rate measured?
  9. Section 2. Case Presentation. In the introduction, the authors have presented that severe COVID-19 infections is complicated with coagulopathy and disseminated intravascular coagulation (DIC) . The use of anticoagulants, preferentially heparin or LMWH, for patients with severe COVID-19 has been recommended by expert consensus and World Health Organization WHO. In addition , the patent presented in this study had a high level of platelets. These data and the clinical observation of the presented patient lead to one question. Why did the authors not switch from warfarin therapy to heparin therapy?
  10. There are also some remarks regarding the relationship between inflammation and clotting, as well as the therapeutic potential of anticoagulants. However, the reviewer came to conclusion that at the present stage of the review, it is first necessary to deal with the problems related directly to case report.
